# PD-L1 Expression in Cutaneous Angiosarcomas: A Systematic Review with Meta-Analysis

Renato Lobrano [1,†], Panagiotis Paliogiannis [1,2,*,†], Angelo Zinellu [3], Giuseppe Palmieri [3,4], Ivana Persico [4], Arduino A. Mangoni [5,6] and Antonio Cossu [1,2]

1   Anatomic Pathology and Histology, University Hospital (AOU) of Sassari, Via Matteotti 60, 07100 Sassari, Italy
2   Department of Medicine, Surgery and Pharmacy, University of Sassari, Viale San Pietro 43, 07100 Sassari, Italy
3   Department of Biomedical Sciences, University of Sassari, Viale San Pietro 43, 07100 Sassari, Italy
4   Institute of Genetic and Biomolecular Research, National Research Council (CNR), Traversa La Crucca 3, 07100 Sassari, Italy
5   Discipline of Clinical Pharmacology, College of Medicine and Public Health, Flinders University, Sturt Road, Bedford Park, Adelaide, SA 5042, Australia
6   Department of Clinical Pharmacology, Flinders Medical Centre, Southern Adelaide Local Health Network, Flinders Drive, Bedford Park, Adelaide, SA 5042, Australia
*   Correspondence: ppaliogiannis@uniss.it
†   These authors contributed equally to this work.

**Abstract:** Cutaneous angiosarcoma (CAS) is the most common type of angiosarcoma that predominantly affects older Caucasians. The outcomes of immunotherapy in CAS are currently under investigation in relation to the expression of programmed death ligand 1 (PD-L1) and other biomarkers. We performed a systematic review and metanalysis of data from the current literature reporting on PD-L1 immunohistochemistry expression. A systematic search of publications in the electronic databases PubMed, Web of Science, and Scopus was conducted using the following terms: "PD-L1" and "angiosarcomas". A total of ten studies reporting on 279 cases were identified and included in the meta-analysis. The pooled prevalence of PD-L1 expression in CAS was 54% (95% CI 36–71%), with high heterogeneity ($I^2$ = 84.81%, $p < 0.001$). In sub-group analysis, the proportion of PD-L1 expression in CAS was significantly ($p = 0.049$) lower in Asian studies (ES = 35%, 95% CI 28–42%, $I^2$ = 0.0%, $p = 0.46$) than in European studies (ES = 71%, 95% CI 51–89%, $I^2$ = 48.91%, $p = 0.12$).

**Keywords:** skin; cancer; cutaneous angiosarcomas; PD-L1; immunohistochemistry; immunotherapy

## 1. Introduction

Angiosarcomas are endothelial tumors that are characterized by high rates of metastases and recurrences. Angiosarcomas can arise in different organs and tissues, particularly in the skin and soft tissues, parenchymatous organs (liver, spleen), and bone [1]. Cutaneous angiosarcoma (CAS), the most common type of angiosarcoma, predominantly affects older Caucasians. The five-year overall survival has been reported to be less than 20% in Japanese patients but substantially higher in Western populations at 31–50% [2]. Surgery, where possible, is the main therapy, whereas traditional chemotherapy with anthracyclines, paclitaxel, docetaxel, and gemcitabine, with or without radiotherapy, is generally used in patients with unresectable and advanced disease. In this setting, novel treatments, e.g., anti-vascular growth factor receptor (VEGF) drugs, eribulin mesylate, and immune checkpoint inhibitors (ICIs), have been recently introduced in clinical practice and research to further improve outcomes [2].

ICIs, immuno-oncologic agents that potentiate T cell-mediated anti-tumor immunity, have represented a major therapeutic advance particularly in advanced stage melanoma, non-small cell lung cancer (NSCLC), head and neck, and breast cancer, in relation with the levels of the programmed-death ligand 1 (PD-L1) expression [3–6]. Recent reports

have highlighted the potential benefit of ICI-based treatment in patients with advanced angiosarcomas [7–10]. This makes the study of the immunohistochemical expression of PD-L1 a critical factor in selecting patients for the effective use of immunotherapy in advanced-stage CAS.

Studies have recently investigated the PD-L1 expression levels in CAS using a range of assays, antibodies, and expression thresholds [8]. The aim of this systematic review and meta-analysis was to critically appraise the current evidence on PD-L1 expression in CAS in order to provide useful information to guide clinical decisions and assist in the design of prospective intervention studies based on this precision medicine approach.

## 2. Materials and Methods

### 2.1. Search Strategy, Eligibility Criteria, and Study Selection

A systematic search of publications in the electronic databases PubMed, Web of Science, and Scopus, from inception to the 12th of July 2022, was conducted using the following terms: "PD-L1" and "angiosarcomas". Abstracts were independently screened by two investigators (PP and RL) to establish relevance. If relevant, full articles were independently reviewed according to the following eligibility criteria: (i) studies including pathologically diagnosed CAS, (ii) studies reporting the immunohistochemistry assessment of PD-L1 expression in CAS irrespective of antibody type, (iii) English language used, (iv) studies with ethical approval performed in accordance with the declaration of Helsinki, and (v) full-text publication available. In vitro and animal studies, expert opinions, commentaries, studies without relevant data on CAS, and reviews without original data were excluded. The references of the retrieved articles were also searched to identify additional studies. Any disagreement between reviewers was resolved by a third investigator (AC). The risk of bias was assessed using the Joanna Briggs Institute (JBI) checklist for prevalence studies. Data on the mean age, sex, publication year, country where the study was conducted, type of antibody used, PD-L1 evaluation scores, thresholds, and number of tested and positive cases, were extracted from each study. The meta-analysis was registered to Prospero (CRD42022348229); The PRISMA guidelines for reporting systematic reviews and meta-analyses were followed.

### 2.2. Statistical Analysis

Forest plots of pool proportion were assessed by using the "metaprop" Stata command by performing a DerSimonian-Laird random-effects meta-analysis of proportions obtained from individual studies [11]. In order to stabilize the variances, the Freeman–Tukey double arcsine transformation was also applied to stabilize the variances in order to calculate the pooled estimates [11]. The $I^2$ statistic was used to assess statistical heterogeneity. $I^2$ values <50% indicated low, 50–75% moderate, and >75% high heterogeneity, respectively [12,13]. In case high heterogeneity was present, a random-effects model was applied. A sensitivity analysis was conducted to investigate the influence of individual studies on the overall risk estimate [14]. In order to evaluate the presence of potential publication bias, the associations between study size and magnitude of effect were analyzed by means of Begg's adjusted rank correlation test and Egger's regression asymmetry test at the $p < 0.05$ level of significance [15,16]. The Duval and Tweedie "trim-and-fill" procedure was performed to further test and eventually correct the presence of publication bias [17]. Univariate meta-regression analyses were conducted to investigate associations between the effect size and the publication year and sample size. Subgroup analyses were also performed according to specific cut-offs and the continent where the study was conducted. Statistical analyses were performed using Stata 14 (Stata Corp., College Station, TX, USA).

## 3. Results

### 3.1. Literature Search and Study Selection

A flow chart describing the screening process is described in Figure 1. We initially identified 90 studies. After the exclusion of duplicates, irrelevant studies, abstracts, reviews,

and other studies not fulfilling the selection criteria, a total of 10 studies reporting on 279 cases were included in the meta-analysis [1,8,10,18–24]. The characteristics of the retrieved studies, published between 2016 and 2022, are presented in Table 1.

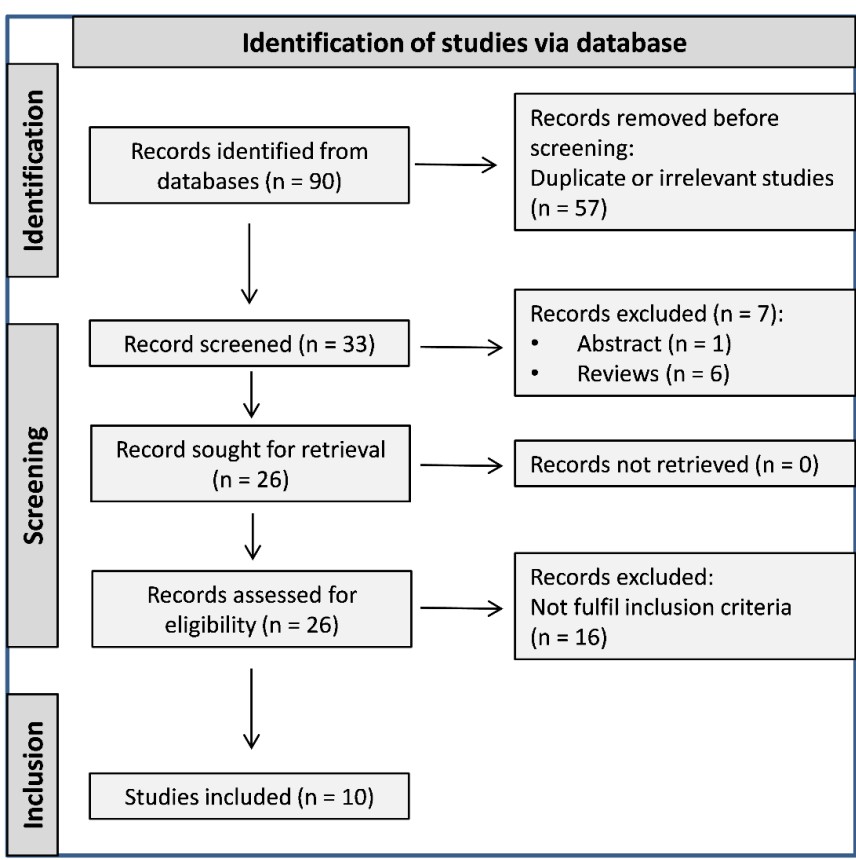

**Figure 1.** Flowchart depicting the study selection process.

**Table 1.** Summary of the studies on PD-L1 expression in subjects with cutaneous angiosarcoma included in the meta-analysis.

| First Author Year, Country | n | Age Mean ± SD | Sex (M/F) | PD-L1 Antibody Clone | Cut-Off | Cut-Off < 1% | Cut-Off 1–49% | Cut-Off > 50% | PD-L1 Positive |
|---|---|---|---|---|---|---|---|---|---|
| Shimizu et al., 2016, Japan [8] | 52 | 76 | 33/19 | NR | >5% | NR | NR | NR | 21 |
| Botti et al., 2017, Italy [1] | 4 | NR | NR | SP142 | >5% | NR | NR | NR | 3 |
| Honda et al., [10] 2017, Japan | 106 | 74,5 | 75/31 | SP142 | >5% | NR | NR | NR | 32 |
| Kawamura et al., 2019, Germany [18] | 29 | 76 | 18/11 | NR | >5% | NR | NR | NR | 22 |
| Gambichler et al., 2020, Germany [19] | 12 | 72 | 8/4 | ab205921 | NR | NR | NR | NR | 5 |
| Okabayshi et al., 2020, Japan [20] | 20 | NR | NR | E1L3N | >5% | NR | NR | NR | 8 |
| Bi et al., 2021, China [21] | 21 | 67 | 14/7 | 22C3 | >1, 1–5, 5–10, 10–50, >50 | 12 | 4 | 5 | 9 |
| Espejo-Freire et al., 2021, USA [22] | 11 | NR | NR | SP142 | >5% | NR | NR | NR | 1 |
| Googe et al., 2021, USA [23] | 10 | 73 | 5/5 | ZR3 | <1, 1–49, >50 | 0 | 3 | 7 | 10 |
| Tomassen et al., 2022, Netherlands [24] | 14 | NR | NR | E1L3N | <1, 1–10, 10–49, >50 | 2 | 11 | 1 | 12 |

F: female; M: male; NR: not reported. Cut-offs are expressed as proportions or ranges of positive, viable tumor cells, substantially reproducing the Tumor Proportion Score (TPS).

### 3.2. PD-L1 Expression in Cutaneous Angiosarcomas: Results of Individual Studies and Syntheses

All studies retrospectively considered the staining of the cell membrane of the tumoral component to evaluate PD-L1 expression. The percentage of stained tumoral cells used as a cut-off value to distinguish between cases with and without PD-L1 expression is reported in Table 1; some studies tested more than one cut-off value. The forest plot for the proportion of PD-L1 immunohistochemistry expression in CAS is reported in Figure 2. Point prevalence ranged from 9% (95% CI 2–38%) [22] to 100% (95% CI 72–100%) [23]. The pooled prevalence of PD-L1 expression in CAS was 54% (95% CI 36–71%), with high heterogeneity ($I^2 = 84.81\%$, $p < 0.001$). Sensitivity analysis showed that the corresponding pooled proportion values were not substantially altered when individual studies were sequentially removed (effect size range, between 49 and 59%, Figure 3).

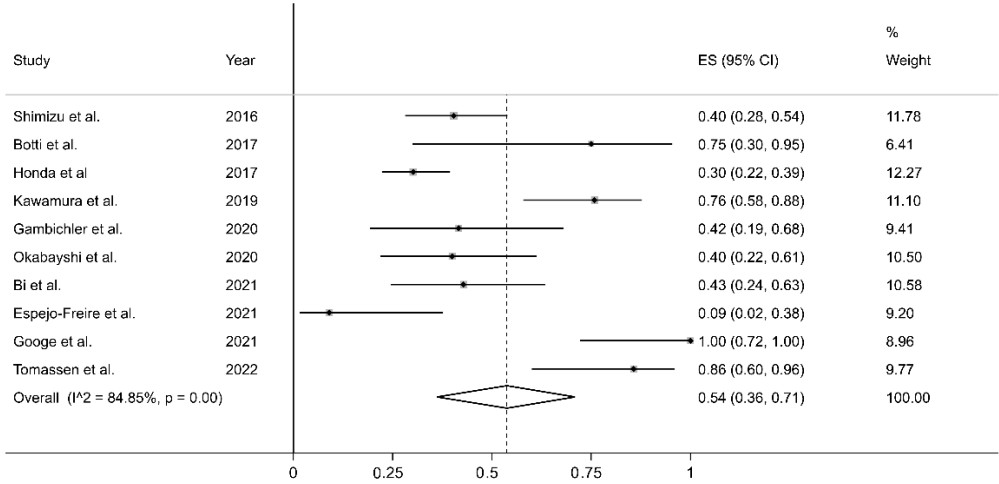

**Figure 2.** Forest plot of studies examining the proportion of PD-L1 expression in CAS. [1,8,10,18–24].

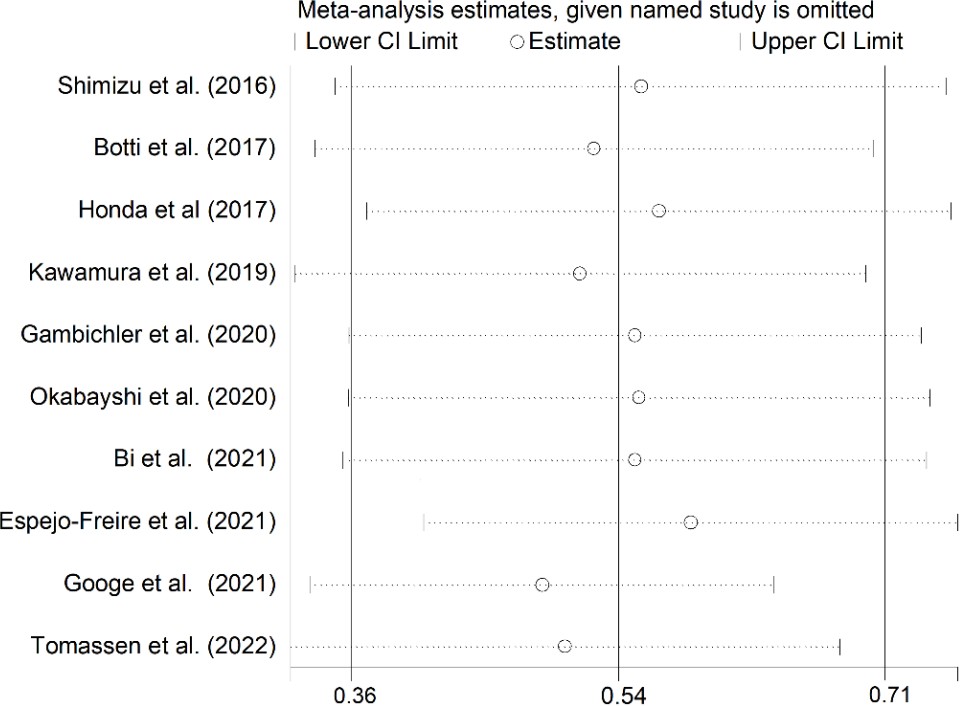

**Figure 3.** Sensitivity analysis of the PD-L1 expression frequencies in CAS. For each study, the displayed effect size (hollow circles) corresponds to an overall effect size computed from a meta-analysis excluding that study [1,8,10,18–24].

### 3.3. Quality Assessment

The risk of bias was considered low in seven studies and high in the remaining three (Supplementary Table S1); considering the small number of studies recruited, we decided to include the three studies with a high risk of bias and to perform a detailed analysis of the publication bias.

### 3.4. Publication Bias

Small-study effects analysis indicated the absence of significant publication bias (Begg's test, *p* = 0.24; Egger's test, *p* = 0.13). However, the "trim-and-fill" method identified four potential missing studies to be added to the left side of the funnel plot to ensure symmetry (Figure 4). The adjusted proportion decreased to 33% (95% CI 11–55%).

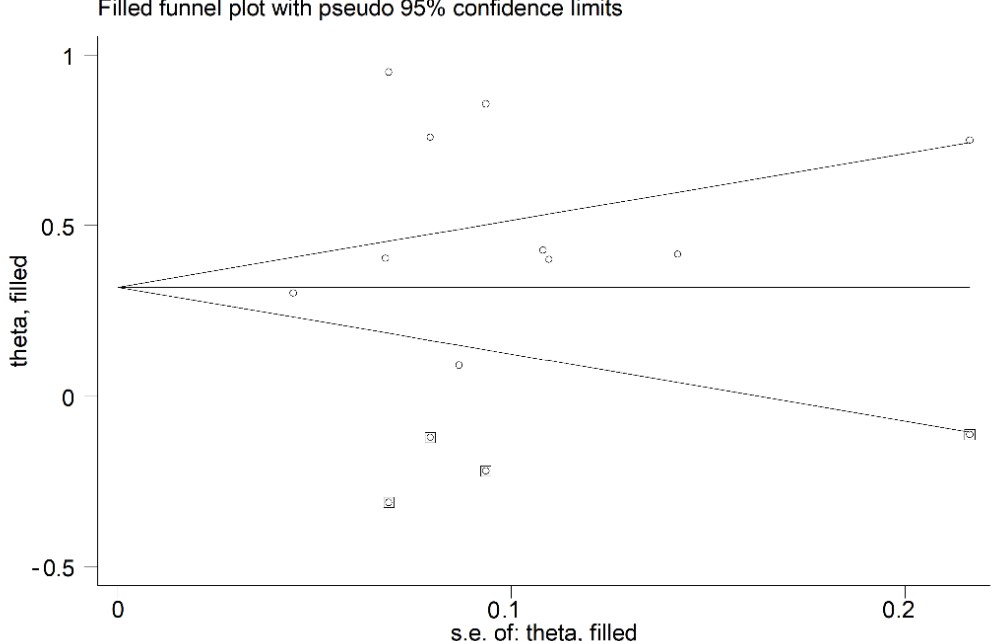

**Figure 4.** Funnel plot of studies investigating PD-L1 expression frequencies in CAS after trimming and filling. Dummy studies and genuine studies are represented by enclosed circles and free circles, respectively.

### 3.5. Meta-Regression and Sub-Group Analysis

In univariate meta-regression, no significant associations were observed between the effect size and publication year (t = 1.25, *p* = 0.246). A trend toward a significant association was observed between the effect size and sample size (t = −1.93, *p* = 0.09, Figure 5). Meta-regression analysis of age, sex and antibody type was not possible due to lacking or poor information in the retrieved studies.

In sub-group analysis, the proportion of PD-L1 positivity in CAS was significantly (*p* = 0.049) lower in Asian studies (ES = 35%, 95% CI 28–42%, $I^2$ = 0.0%, *p* = 0.46) than in European studies (ES = 71%, 95% CI 51–89%, $I^2$ = 48.91%, *p* = 0.12, Figure 6). The between-study variance was significantly reduced in European studies and virtually absent in Asian studies. No significant differences in effect size were observed between studies conducted in Asia and America (*p* = 0.534) or between America and Europe (*p* = 0.612). Non-significantly lower proportions (*p* = 0.16) of PD-L1 positivity were reported using a cut-off >5% (ES = 42%, 95% CI 25–61%, $I^2$ = 81.14%, *p* = 0.001) when compared to other cut-offs (ES = 80%, 95% CI 38–100%, *p* = 0.001, Figure 7).

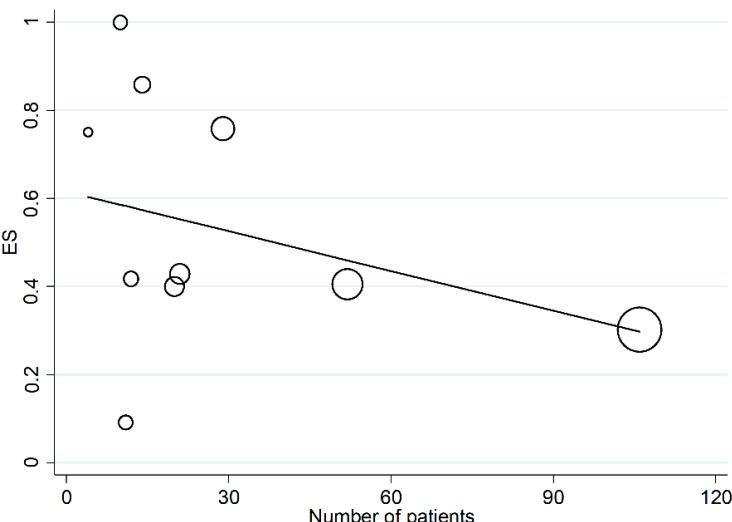

**Figure 5.** Forest plot of studies examining the proportion of PD-L1 expression in CAS according to the continent where studies were conducted. The *p*-values were not assessable in very small subgroups [1,8,10,18–24].

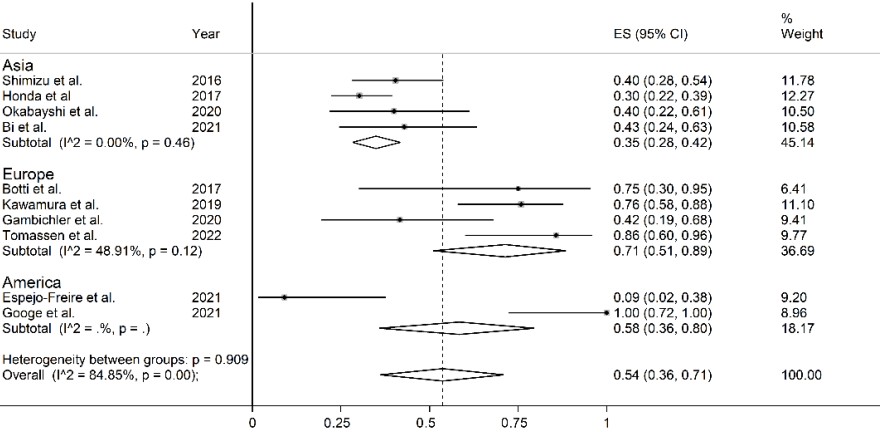

**Figure 6.** Meta-regression analysis showing the correlation between effect size and number of recruited patients.

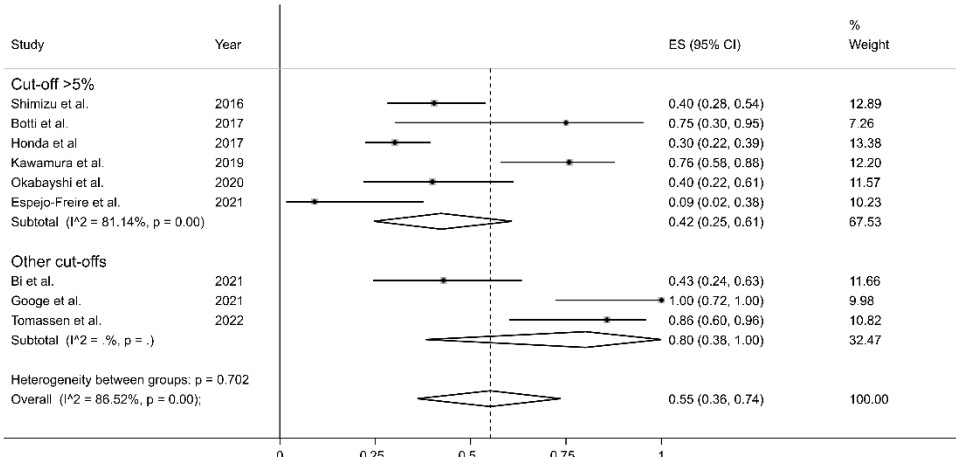

**Figure 7.** Forest plot of studies examining the proportion of PD-L1 expression in CAS according to cut-off values employed. The *p*-values were not assessable in very small subgroups [1,8,10,18–24].

## 4. Discussion

Immunotherapy is currently in use for the treatment of several solid cancers, like NSCLC, melanoma, head and neck, colorectal, and breast cancer [3–6], and represents a promising therapeutic option in patients with advanced CAS. Immunotherapy agents like ICIs act on the immune system of the patients, modifying the tumoral microenvironment and unlocking defensive functions which are locked through the expression of specific antigens by tumor cells. Nevertheless, not all patients benefit from this mechanism, as only a variable fraction of them, depending on the tumor type, responds to ICIs. For this reason, specific biomarkers have been identified for the selection of patients to submit to immunotherapy, and the efficacy of immune-related medications in solid tumors is based on the expression of specific biomarkers, e.g., PD-L1 and Cytotoxic T-Lymphocyte Antigen 4 (CTLA-4), which indicate the tumor's susceptibility to this class of drugs. In particular, the expression levels of PD-L1 are increasingly considered useful to facilitate a precision medicine approach in several solid cancers.

PD-L1 is a protein composed of 290-amino acids, belonging to the B7 family of type I transmembrane protein receptors, which are characterized by two extracellular functional structures (the IgV-like and IgC-like domains), a transmembrane domain and an intracellular cytoplasmic domain. PD-L1 is expressed in several immune cell types, including antigen-presenting cells, T cells, B cells, monocytes, and epithelial cells [25]. After activation by specific proinflammatory cytokines, these cells increase the expression of PD-1, another effector of the same family which represents the natural binding counterpart of PD-L1 [26]. The binding of PD-L1 to PD-1 activates the downstream signaling of PD-1 receptor in T cells, inhibiting their proliferation, as well as cytokine generation and release, which finally leads to T cell cytotoxicity blockade. This occurs because the physiological role of immune checkpoints is to prevent and attenuate excessive immune attacks on self-antigens during immune responses and inflammation [26]. Numerous solid tumors take advantage of these mechanisms to overcome the host's immune defense, survive, grow, and progress. Enhancing the expression of checkpoint inhibitory proteins, like PD-1 and PD-L1, reduces anti-tumor immune responses, allowing cancer cell survival and metastasis. On the other hand, these molecular processes are the targets of modern ICIs, and the immunohistochemical detection of these proteins, especially that of PD-L1, is currently used for the selection of patients for immunotherapy with ICIs, as mentioned before.

Ipilimumab, a CTLA4 inhibitor, was the first ICI approved by the U.S. Food and Drug Administration (FDA) in 2011 for the treatment of advanced-stage melanoma. A few years later, in 2014, it was also approved as the first anti-PD-1 antibody, pembrolizumab, for use in metastatic melanoma [25]. Subsequently, FDA and other international and national regulatory institutions have approved additional ICIs, some of them blocking PD1 (i.e., Pembrolizumab, Nivolumab, Cemiplimab, etc.), and others PD-L1 (i.e., Atezolizumab, Avelumab, Durvalumab, etc.). In addition, a great number of checkpoint-inhibiting monoclonal antibodies are currently under investigation in numerous ongoing clinical trials. This is because of the great success of immunotherapy in increasing progression-free and overall survival in patients with several types of advanced-stage cancers. For example, in NSCLC, several ICIs are currently used in all treatment lines in PD-L1 expressing cases, with unquestionable survival advantages certified in both clinical trials [27,28] and large real-life studies; in a recent nationwide real-world study performed in Denmark, the authors found that three-year overall survival tripled from 6% to 18% after implementation of immunotherapy [29]. In addition, it is currently estimated that long-term survival can be achieved in more than 15% of advanced-stage NSCLC patients treated with immunotherapy, with some authors suggesting treatment discontinuation after two years of disease stability [30].

Recent studies have investigated the association between PD-L1 and clinical responses to immunotherapy in CAS. Lee et al. reported that PD-L1 status is an independent prognostic factor for overall survival in metastatic angiosarcoma patients, including those with CAS [7]. Bi et al. did not find any statistically significant correlation between PD-L1 expres-

sion and survival in a cohort of surgically resected patients [21]. Honda et al. found that PD-L1 expression had a prognostic significance only when combined with that of PD-1 [10]. Finally, Shimitzu et al. reported that PD-L1 expression predicts poor outcomes in a cohort of 52 patients with CAS [8]. Similar results were also reported by Orth et al. in patients with soft tissue sarcomas, including CAS [9]. Whilst generally, these studies support the presence of a significant association between PD-L1 expression and outcomes in CAS, there was a significant heterogeneity in experimental design, retrospective vs. prospective, medications used and treatment protocols.

PD-L1 expression is generally determined by immunohistochemistry, with established antibodies, platforms, procedures and interpretation methods. Currently, four PD-L1 immunohistochemical assays registered with the FDA with four different PD-L1 antibodies (22C3, 28–8, SP263, SP142) on two different platforms (Dako and Ventana), each with a specified scoring system [25]. The Tumor Proportion Score (TPS) evaluates the proportion of tumor cells with membrane staining for PD-L1 among the total number of viable tumor cells. The Combined Positive Score (CPS) is a scoring system that calculates the fraction of both tumor and immune (lymphocytes and macrophages) cells expressing PD-L1 divided by the total number of tumor-viable cells. A pathological evaluation must be performed with a median magnification (20x), and at least 100 tumoral viable cells need to be detected for the evaluation of both scoring systems. TPS is used for PD-L1 expression definition in NSCLC, with different ICIs employed in different treatment lines in patients with PD-L1 between 1% and 49% and in patients with PD-L1 greater than 50%. CPS is generally used in breast, gastrointestinal and cervix cancer, with different cut-offs in different tumor types [31].

The lack of specific guidelines for the pathological evaluation of PD-L1 in CAS has led to the use of a wide range of assays, antibodies, and cut-offs, as depicted in Table 1. The studies identified in this meta-analysis used five different antibodies. However, in each case, the positivity was judged with respect to the tumoral cell membrane staining, substantially reproducing the TPS; few authors evaluated staining in tumor-infiltrating cells (TILs) with different cut-offs, but they were not analyzable due to their low number and heterogeneity. Most authors selected a 5% cut-off level; only three of them used more than one threshold. These differences represent a substantial confounding factor that may account, at least in part, for the high heterogeneity observed in our analysis. However, the pooled PD-L1 expression prevalence of 54% was not substantially altered when individual studies were sequentially removed (Figure 3). Furthermore, no significant publication bias was observed. The results of our meta-analysis suggest that approximately half of CAS cases might be particularly suitable for treatment with immune-modulating drugs.

Interesting clues emerged from meta-regression and subgroup analyses. In particular, a trend toward a significant association between the effect and sample size was observed, highlighting the need for future studies with greater cohorts. In sub-group analysis, the proportion of PD-L1 expression in CAS was significantly lower in Asian than European cohorts (Figure 5); in addition, heterogeneity was virtually removed in studies conducted in Asian patients, suggesting the presence of ethnic-related differences in PD-L1 expression. Furthermore, the two studies published in 2021 in the USA showed extremely discordant results, with the prevalence of PD-L1 expression ranging from 9% (95% CI 2–38%) [22] to 100% (95% CI 72–100%) [23]. The lower expression levels of PD-L1 in Asian cohorts are particularly interesting, considering the lower survival rates reported in these patients, as previously described. These issues notwithstanding, an overall greater standardization in methodological approaches is warranted in future studies.

## 5. Conclusions

Our metanalysis showed that the pooled prevalence of PD-L1 expression in CAS is 54%. High heterogeneity was observed among the included studies, especially those performed in Western countries. Prospectively designed studies with larger cohorts and standardized laboratory approaches are necessary to further confirm the potential role of

assessing the levels of PD-L1 expression for the selection of immunotherapeutic strategies in patients with advanced CAS.

**Supplementary Materials:** The following supporting information can be downloaded at: https://www.mdpi.com/article/10.3390/curroncol30050388/s1, Table S1: The Joanna Briggs Institute critical appraisal checklist.

**Author Contributions:** Conceptualization, R.L. and P.P.; methodology, A.Z. and A.A.M.; software, A.Z.; formal analysis, A.Z. and A.A.M.; investigation, R.L., P.P., G.P. and I.P.; resources, R.L., P.P., G.P. and I.P.; data curation, A.Z. and A.A.M.; writing—original draft preparation, R.L., P.P., G.P. and I.P.; writing—review and editing, A.A.M. and A.C.; supervision, R.L. and A.C.; funding acquisition, A.C. All authors have read and agreed to the published version of the manuscript.

**Funding:** A.C. received research funds from the "Fondo di Ateneo per la ricercar 2020", University of Sassari.

**Conflicts of Interest:** The authors declare no conflict of interest.

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
