# Peer review of "PD-L1 Expression in Cutaneous Angiosarcomas: A Systematic Review with Meta-Analysis"

_curroncol, doi:10.3390/curroncol30050388_

Round 1

Reviewer 1 Report

The manuscript titled "PD-L1 expression in cutaneous angiosarcomas: a systematic re-2 view with metanalysis" is interesting and can be considered for publication in Current Oncology. The article published between which is considered for the review in this manuscript - the details can be included. The result and discussion are well presented. Overall this manuscript is well presented and can be considered for publication in current oncology.

Author Response

The authors wish to thank the reviewer for his/her revision. 

Reviewer 2 Report

1. Please clarify the inclusion criteria. One of the criteria is "studies reporting the immunohistochemistry assessment of PD-L1 expression in CAS with commercially available antibodies", but in table 1, the clone of the antibodies were not reported in some of the recruited studies.

2. Again in table 1, please explain the meaning of the different forms of cutoff

3. Some p values are mising in figure 5 and 6

4. Indeed, as stated by the authors in the last 3 paragraphs of discussion, the methods used to evaluate of PD-L1 expression in CAS incouding different assays, antibodies, and cut-offs, could influence the conclusions of the selected studies. Could these be the reasons resulting in the differences seen between Europeans and Asians? 

Author Response

Dear reviewer

We would like to thank you for your suggestions; corrections are highlighted in yellow in the text and summarized here point by point:

Issue 1. Please clarify the inclusion criteria. One of the criteria is "studies reporting the immunohistochemistry assessment of PD-L1 expression in CAS with commercially available antibodies", but in table 1, the clone of the antibodies were not reported in some of the recruited studies.

Reply 1. Initially we aimed to include papers including only commercially available antibodies, but the number of articles was very low. For this reason we decided to include papers irrespective of antibody type, but we forgot to change the corresponding sentence in the inclusion criteria. The sentence was now corrected.

Issue 2. Again in table 1, please explain the meaning of the different forms of cutoff.

Reply 2. A sentence to better explain the cut-off values reported in Table 1 has been added in the 3.2 section of the Results. Further explanations on the use of cut-off values are included in the Discussion section. 

Issue 3. Some p values are mising in figure 5 and 6

Reply 3. Meta-analysis of proportion was normally performed by metan command. However the use of this command did not allow entry of studies with 0% or 100% proportions that were excluded from the meta-analysis. Then we use metaprop command (see doi: 10.1186/2049-3258-72-39 ) that allows inclusion of studies with proportions equal to zero or 100 percent (like study of Googe et al. reference 27) and avoids confidence intervals exceeding the 0 to 1 range. However, when analysis was restricted to a really small number of studies like sub-group analysis of figure 6 (sub-group of American studies, only two studies) or figure 7 (sub-group of other cut-offs, only three studies) also the metaprop command was not able to perform heterogeneity evaluation in these sub-group, due to the presence of a study with 100% proportion. For this reasons some I2 and p-value were missing. We added a sentence to underly this in the corresponding figure legends.

Issue 4. Indeed, as stated by the authors in the last 3 paragraphs of discussion, the methods used to evaluate of PD-L1 expression in CAS incouding different assays, antibodies, and cut-offs, could influence the conclusions of the selected studies. Could these be the reasons resulting in the differences seen between Europeans and Asians? 

Reply 4. The difference between Europeans and Asians probably is not due to these factors, because they were randomly distributed among them. Nevertheless, subgroup analyses were not possible considering the small number of cases. We stated that in the 3.5 section of the Results.   

Reviewer 3 Report

Brief Overview of the manuscript:

Cutaneous angiosarcoma, the most prevalent type of angiosarcoma, is commonly observed in elderly Caucasian individuals. To evaluate the prevalence of PD-L1 expression in CAS, the authors conducted a systematic review and meta-analysis. The analysis included ten studies and 279 cases, and the overall prevalence of PD-L1 expression in CAS was found to be 54%, with substantial variability. Furthermore, a subgroup analysis indicated that the frequency of PD-L1 expression was significantly lower in studies conducted in Asia compared to those conducted in Europe.

Major Points:

1.     In section 3.3 (Quality assessment), the authors should provide a detailed explanation of the risk of bias analysis presented in supplemental table 1, as well as clarify their rationale for including studies with a high risk of bias in the meta-analysis.

2.     It would be valuable if the authors perform an analysis on overall patient survival versus PD-L1 expression, where relevant data is available. However, the authors should acknowledge that the treatments received by the patients could potentially skew this meta-analysis.

3.     The authors could also perform a meta-regression analysis of age and sex by including the studies where corresponding information is available. This would provide additional insights into the potential impact of these factors on the meta-analysis results.

Minor Points:

1.     In line 3 (Title), please correct the spelling of "metanalysis" to "meta-analysis."

2.     In line 51, please provide the full form of the word "TKI."

3.     In table 1, please define the term "NR" in the table legend.

4.     In table 1, please define the cut-off used and clarify if it is based on TPS or CPS.

5.     In figures 6 and 7, please include the level of heterogeneity (I2) and statistical significance value (P) for the American group and other cut-offs, respectively.

6.     In line 207, please change the word "CTL4" to "CTLA4."

7.     In line 220, please rephrase the sentence to avoid repeating the phrase "found that" twice.

Author Response

Dear reviewer

We would like to thank you for your suggestions; corrections are highlighted in yellow in the text and summarized here point by point:

Major Points:

Issue 1. In section 3.3 (Quality assessment), the authors should provide a detailed explanation of the risk of bias analysis presented in supplemental table 1, as well as clarify their rationale for including studies with a high risk of bias in the meta-analysis.

Reply 1. The main reason we chose to include paper with high risk of bias is the small number of papers found in the literature. We preferred to include them all and report data on quality assessment. This allowed some calculations which were impossible to make with a small number of papers to be made. We stated that better in the quality assessment section.  

Issue 2. It would be valuable if the authors perform an analysis on overall patient survival versus PD-L1 expression, where relevant data is available. However, the authors should acknowledge that the treatments received by the patients could potentially skew this meta-analysis.

Reply 2. Our work is a pathology study focusing only on the immunohistochemical expression of PD-L1 in CAS. The use of PD-L1 expression as a biomarker for several therapies is a medical oncology issue that goes over the aims of our study.

Issue 3. The authors could also perform a meta-regression analysis of age and sex by including the studies where corresponding information is available. This would provide additional insights into the potential impact of these factors on the meta-analysis results.

Reply 3. The meta-regressions you suggest gave the following results:

Age (t=0.19; p=0.859)

Ratio M/F (t=-2.47; p=0.069)

Nevertheless, we chose to not report them because according to Cochrane guidelines “meta-regression should generally not be considered when there are fewer than ten studies in a meta-analysis”. For this reason, we wrote “Meta-regression analysis of age and sex was not possible due to the paucity of information in the retrieved studies” in the text. In any case, if you believe that these data should be added, we can add them in a subsequent revision.

Minor Points:

Issue 1. In line 3 (Title), please correct the spelling of "metanalysis" to "meta-analysis."

Reply 1. The title was modified as suggested.

Issue 2. In line 51, please provide the full form of the word "TKI."

Reply 2. This was an error, TKI was replaced with ICI.

Issue 3. In table 1, please define the term "NR" in the table legend.

Reply 3. Table legend explaining all the abbreviations has been added.

Issue 4. In table 1, please define the cut-off used and clarify if it is based on TPS or CPS.

Reply 4. Details were added in figure legend. Further details regarding the scores used in the meta-analysis are included in the discussion.

Issue 5. In figures 6 and 7, please include the level of heterogeneity (I2) and statistical significance value (P) for the American group and other cut-offs, respectively.

Reply 5. Meta-analysis of proportion was normally performed by metan command. However the use of this command did not allow entry of studies with 0% or 100% proportions that were excluded from the meta-analysis. Then we use metaprop command (see doi: 10.1186/2049-3258-72-39 ) that allows inclusion of studies with proportions equal to zero or 100 percent (like study of Googe et al. reference 27) and avoids confidence intervals exceeding the 0 to 1 range. However, when analysis was restricted to a really small number of studies like sub-group analysis of figure 6 (sub-group of American studies, only two studies) or figure 7 (sub-group of other cut-offs, only three studies) also the metaprop command was not able to perform heterogeneity evaluation in these sub-group, due to the presence of a study with 100% proportion. For these reasons some I2 and p-value were missing. We added a sentence to underly this in the corresponding figure legends.

Issue 6. In line 207, please change the word "CTL4" to "CTLA4."

Reply 6. The error was corrected.

Issue 7. In line 220, please rephrase the sentence to avoid repeating the phrase "found that" twice.

Reply 7. The error was corrected.

English language and style of the article was revised by an academic author expert in English medical writing (Prof. Arduino Aleksander Mangoni, University of Adelaide, Australia). 

Round 2

Reviewer 2 Report

The authors have made the corrections, no further comments

Reviewer 3 Report

Thank you for the revised manuscript.